# ParRouting: An Efficient Area Partition-Based Congestion-Aware Routing Algorithm for NoCs

**DOI:** 10.3390/mi11121034

**Published:** 2020-11-25

**Authors:** Juan Fang, Di Zhang, Xiaqing Li

**Affiliations:** Faculty of Information Technology, Beijing University of Technology, Beijing 100124, China; zhangd@emails.bjut.edu.cn

**Keywords:** congestion-aware, adaptive routing, load-balance, networks-on-chip

## Abstract

Routing algorithms is a key factor that determines the performance of NoC (Networks-on-Chip) systems. Regional congestion awareness routing algorithms have shown great potential in improving the performance of NoC. However, it incurs a significant queuing latency when practitioners use existing regional congestion awareness routing algorithms to make routing decisions, thus degrading the performance of NoC. In this paper, we propose an efficient area partition-based congestion-aware routing algorithm, ParRouting, which aims at increasing the throughput and reducing the latency for NoC systems. First, ParRouting partitions the network into two areas (i.e., edge area and central area.) based on node priorities. Then, for the edge area, ParRouting selects the output node based on different priorities for higher throughput; for the central area, ParRouting selects the node in the low congestion direction as the output node for lower queuing latency. Our experimental results indicate that ParRouting achieves a 53.4% reduction in packet average latency over SPLASH -2 ocean application and improves the saturated throughput by up to 38.81% over a synthetic traffic pattern for an NoC system, compared to existing routing algorithms.

## 1. Introduction

Network-on-chip (NoC) offers a flexible and scalable solution to satisfy the communication bandwidth for the many-core system-on-chip (SoC) [1,2,3,4,5]. Suffering from high power cost, long network latency, and low network throughput [2,6,7], the overall performance of existing NoC systems is undesirable. Many approaches have been proposed to improve NoC performance, which includes changing the NoC structure [7,8], dynamically changing packet injection rate [9,10], optimizing routing algorithms [5,11], and so on. Routing algorithm optimization is one of a lower cost solution [12,13,14,15,16,17,18]. Routing algorithms specify the transmitted path of a packet on NoC systems, and thus can directly influence the throughput and latency of the NoC [11,19]. Deterministic routing algorithms are first introduced to improve the performance of NoC in the early days [20,21]. Unfortunately, they fail to efficiently balance the network load since they do not consider the dynamic network status during the routing process [22]. To cope with this problem, adaptive routing algorithms are proposed [4,23] recently, so as to reduce the latency and increase the throughput of the NoC at a small cost [2,24]. By leveraging the dynamic network status during the routing processes, they have shown great potential and can be widely applied in interconnection networks.

From the perspective of congestion-aware routing, adaptive routing algorithms can be classified into three types: (1) Local congestion awareness: Its key idea is to select router output ports by only considering the congestion status of neighboring nodes [25]. As a result, it usually destroys the global load balance, and thus cannot effectively solve the congestion problem. (2) Global congestion awareness: Its key idea is to maintain fine-grained congestion information of the network. For a given “source-destination” pair, it accurately calculates the amount of congestion on each candidate path of the packet for the minimum path range [12,13]. Different from local congestion awareness algorithms, it considers the global load balance during the routing process and thus can specify the best path to the destination in theory. However, it is a high-cost solution for an NoC system. The main reason is that calculating each candidate path requires a significant amount of computing time, thus increasing the network delays. (3) Regional congestion awareness: Its key idea is to collect congestion information from remote links and weigh congestion values by distance from the local node. Compared with local congestion awareness algorithms, each node in regional congestion awareness algorithms has more information to make routing decisions, thus delivering a lower network latency. Generally, considering a better load balance and lower latency of the NoC, regional congestion awareness algorithms have been proven to be an effective solution, and thus have been the most common congestion awareness routing algorithm.

However, it incurs a significant queuing latency when practitioners use existing regional congestion awareness routing algorithms to select the best path, thus leading to an undesirable performance of the NoC. There are two main reasons behind the above performance problem. First, existing regional congestion awareness algorithms make routing decisions by evaluating the redundant remote link congestion information. When the source node and target node are very close, these remote links are not in the range of the optional path from the source node to the target node. Considering that this information may be wrong, the link that has congested neighbors but is idle at the remote end is selected as the output link. Instead, it interferes with routing decisions and increases queuing delay. Second, the load of the nodes in the central area is significantly higher than the load of the edge nodes under a high traffic injection rate. Existing routing algorithms do not consider shifting the intermediate load to the edge of the network. Therefore, existing regional congestion awareness algorithms can not meet the performance requirements of the NoC yet, and it is non-trivial to optimize the regional congestion awareness algorithm further.

To this end, we propose an efficient regional congestion awareness routing algorithm, ParRouting, which aims at increasing the throughput and reducing queuing latency of NoC. First, ParRouting divides the network nodes into three priorities based on link utilization. Second, according to the nodes’ position, ParRouting sets the nodes with high and medium priority as edge nodes, and the low priority as central nodes. Finally, for edge nodes, ParRouting selects the output node by preferring the high-priority candidate link, so as to better balance the network load. For central nodes, ParRouting selects the output node by evaluating the hotspot status of the remote links and the number of candidate nodes’ free VCs, so as to avoid the congestion area and reduce queuing latency. The goal of ParRouting is to balance the traffic load and reduce the queuing latency of packets, thus improving the overall performance of the NoC systems further.

We evaluate the performance of ParRouting by comparing it with three typical routing algorithms over synthetic traffic patterns and real application patterns. Compared with exiting routing algorithms, ParRouting can better balance the traffic load and yield lower queuing latency of packets over different routing scenarios. Our experimental results indicate that ParRouting achieves a 44.9% reduction in packet average latency over SPLASH-2 water application and improves the saturated throughput by up to 28.72% over a synthetic traffic pattern for an NoC system, compared with existing routing algorithms.

The technical contribution of this paper is three-fold:We propose an efficient regional congestion awareness routing algorithm ParRouting, which can balance the traffic load and reduce the queuing latency of packets for NoC systems;We leverage a new congestion propagation technique, which uses packets to carry congestion information, avoiding the wiring overhead required to propagate congestion information;We conduct comprehensive performance evaluation and analysis on ParRouting over both a synthetic traffic pattern and real application pattern by comparing ParRouting with three typical routing algorithms.

The rest of this paper is structured as follows. Section 2 introduces the related work and motivation. Section 3 presents the design of ParRouting. Section 4 shows the implementation of ParRouting. Section 5 describes the experiment setup and performance evaluation results. Finally, in Section 6, a conclusion is drawn.

## 2. Related Work and Motivation

### 2.1. Related Work

Existing congestion-aware routing algorithms mainly include three types: Local congestion awareness, regional congestion awareness, and global congestion awareness.

For local congestion-aware routing algorithms, routing decisions are made only through the congestion information of the neighboring nodes of the source node, such as DyXY [20] and DyAD [21]. Due to a lack of congestion of the remote link, local congestion awareness may lead to incorrect routing decisions. Therefore, regional congestion-aware algorithms and global congestion-aware algorithms are proposed, in which each node has more congestion information to make routing decisions.

The earliest regional congestion-aware algorithm is RCA [14], which was proposed by Gratz et al. in 2008. This algorithm uses a congestion propagation network to propagate local and remote network status, thereby avoiding network congestion. When calculating the output port, RCA simply considers the status of all routers in the same direction, but in fact, most routing paths do not cross these nodes. The redundant information will interfere with the accuracy of the evaluation of the network congestion status, resulting in the decline of network performance. To solve these problems, Ma Sheng et al. [15] proposed the DBAR routing algorithm based on the RCA algorithm. The DBAR algorithm integrates the target location of the packet into the output port selection, and only considers the state of the node that the packet may pass through, thereby eliminating the redundant network state information. Liu S et al. [16] proposed FreeRider based on DBAR while adopting a new way to propagate congestion. In FreeRider, the congestion value is embedded in the head flit and propagated to downstream nodes with the packet. Ebrahimi et al. [17] proposed the CATRA to achieve a lower packet latency than DBAR through a more additional wiring and proxy nodes, considering the congestion status of a more reasonable node. Reza Akbar et al. [18] proposed ZRA and present a new way for transmitting congestion information which evaluates the probability of passing the relevant nodes and transmit congestion information through a zigzag path. This algorithm performs at a lower latency and has better load balancing but at the cost of higher power consumption.

DAR [26] is an earlier global congestion-aware routing algorithm. In DAR, every node estimates the delay to every other node in the network, and routing decisions are based on these per-destination delay estimates. After DAR, Ramakrishna et al. [12] proposed GCA, which is also a global congestion-aware routing algorithm. Compared to DAR, GCA removes the need for the congestion-propagation network by embedding status information in packet headers. CFPA [22] is a congestion-aware routing that considered the impact of process variation (PV) on message delay. It maintains two routing tables to store multiple paths to every destination via all polar directions. It also provides a full analysis of the different delay components which the algorithm deals with. The congestion information used by the CFPA algorithm for routing decisions has a high resolution, which is useful for fault-tolerant but maybe not necessary for congestion awareness.

### 2.2. Motivation

NoC can provide a flexible and scalable solution to meet the communication requirements for the SoC systems. Optimizing a routing algorithm is the most efficient solution. In particular, regional congestion awareness algorithms have been proven to be an effective solution to improve the performance of NoC systems, and thus has been the most common congestion awareness routing algorithm. However, using existing regional congestion awareness algorithms to make a routing decision can bring in a significant queuing latency, thus degrading the performance of NoC systems. Therefore, existing regional congestion awareness algorithms can not meet the performance requirement of a NoC system yet, and it is essential to optimize existing regional congestion awareness algorithms further so as to increase the throughput and latency of the packets of NoC.

## 3. ParRouting Design

ParRouting is a congestion-aware and load-balance routing algorithm. It has two core ideas. One is to partition the network nodes into three priorities according to the centrality of nodes in the network, then select different routing strategies for nodes in different areas. The other is to use remote congestion information to select the output node.

### 3.1. Network Partition

#### 3.1.1. Closeness Centrality

Centrality is a concept commonly used in social network analysis (SNA) to measures how much a node is in the “center” of other “point pairs” on the way. According to different application scenarios, centrality has four indicators, among which closeness centrality reflects the proximity of a point to other points. It calculates the sum of the shortest distance from a node to all other reachable nodes. After normalization, we get a number between (0, 1). The larger the number, the higher the closeness centrality of this point. A point with a high closeness centrality means that this point is relatively close to any other point, which is at the center of the network. For 2D Mesh topology, nodes are connected to a different number of neighboring nodes. There are obvious differences in the close centrality of different nodes. For 2D Torus topology, all nodes are connected to four other nodes. However, the border nodes are connected by a long link, so that different links can be assigned with different weights to calculate closeness centrality.

The standardized closeness centrality calculation formula is as follows. Closeness centrality of a node *u* is reciprocal of the average shortest path distance to *u* over all n−1 reachable nodes.
(1)$$C\left(u\right)=\frac{n-1}{\sum_{v=1}^{n-1}d\left(v,u\right)}$$
where d(v,u) is the shortest-path distance between *v* and *u*, and *n* is the number of nodes that can reach *u*. For 2D Mesh topology, we simply use 1 to represent the link length of every two adjacent nodes since the link lengths of adjacent nodes are the same. The closeness centrality of an 8×8 2D Mesh network is shown in Figure 1. Each small grid in the figure represents a network node, the value in the small grid is the standardized closeness centrality of the node, and the coordinate axis represents the position of the node in the network. As seen, the closeness centrality of different nodes in the network is obviously different, maxC(u) is 1.75 times of minC(u). We will prioritize nodes based on these values in the next subsection.

#### 3.1.2. Priority and Area Partition

Closeness centrality is a concept related to the shortest path. Most NoC routing algorithms are based on minimum routing. Therefore, closeness centrality is very suitable for predicting the congestion of NoC. Priority partition is the core idea of ParRouting. We partition the priority of the nodes according to their closeness centrality.

The red to blue in Figure 1 represents the closeness centrality of the nodes from high to low. For a 2D Mesh network, the nodes at the center of the network have higher closeness centrality, which means that the shortest distance between these nodes and other nodes is small so that they are more likely to be chosen as output nodes calculated by minimal routing. Therefore, the nodes with a high closeness centrality should be given low priority to reducing the probability of being used, so as to reduce the load of these nodes. Accordingly, the nodes at the network’s edge have a lower closeness centrality so that these nodes should be given high priority to increase the probability of being used. That is, high closeness centrality corresponds to low priority and low closeness centrality corresponds to high priority.

We need two thresholds for the closeness centrality to divide these nodes into three priorities. The smaller one $threshold_{1}$ is the threshold of high priority and medium priority, and the larger one $threshold_{2}$ is the threshold of medium priority and low priority. The thresholds can be formulated as Equation (Equation 2) [18].
(2)$$threshold_{i}=k_{i}\left[maxC(u)-minC(u)\right]+minC(u),0<k<1$$

$k_{1}$ and $k_{2}$ are needed to calculate $threshold_{1}$ and $threshold_{2}$. By following the setting of [18], we assign 0.7 to $k_{2}$, so as to get $threshold_{2}$. Nodes with a closeness centrality greater than $threshold_{2}$ are given low priority, as the red nodes shown in Figure 2. If $k_{2}$ were large, the low priority nodes would be less, which would lead to a smaller central area. So that the remote-congestion-based routing strategy applied in the central area can not work as expected. On the other hand, if $k_{2}$ were small, the central area would be bigger, the priority-based routing strategy applied in the edge area will not work efficiently.

$k_{1}$ is used to get $threshold_{1}$. Nodes with a closeness centrality less than $threshold_{1}$ are given high priority, as the blue nodes shown in Figure 2. If $k_{1}$ were too big or too small, the distinction between high priority nodes and medium priority nodes would reduce, and the load balancing performance would not improve greatly compared to other routing algorithms. Thus, we assign 0.5 to $k_{1}$ to calculate $threshold_{1}$. The rest nodes whose closeness centrality are greater than $threshold_{1}$ and less than $threshold_{2}$ are given medium priority, as the pink nodes shown in Figure 2.

The priority has two uses. 1. Partition the network areas. We apply different routing strategies for different areas. Noticed that the high and medium priority nodes are at the edge of the network and the high priority nodes are at the center of the network. We partition the low priority nodes to the central area. For nodes in the central area, we apply remote-congestion-based routing. We partition the high and medium priority nodes to the edge area. For nodes in the edge area, we apply priority-based routing. 2. Making routing decisions. Priority-based routing strategy evaluates the candidate output node priority to select the output node. We will introduce the two routing strategies in the next subsection.

### 3.2. ParRouting Strategies

ParRouting is a congestion-aware and load-balance routing algorithm, which has two routing strategies for nodes in the edge area and central area respectively. If the source node is in the edge area, we apply priority-based routing otherwise, we apply remote-congestion-based routing.

**Priority-based routing.** If the source node is in the edge area, we make routing decisions according to the node priority. First, the priority of the two candidate output nodes are checked. Second, we must check if the two candidates have free virtual channels (VC). (1) If the two candidates are at the same priority, there are two possibilities for their free VCs: (a) They both have or do not have a free VC. (b) One has a free VC and another does not have a free VC. For situation (a), select an output node randomly and for situation (b), select one that has free VCs as the output node. (2) If the two candidates are at a different priority, there are three possibilities for thei free VCs: (a) They both have or do not have free VCs. (b) The higher priority node has a free VC. (c) The lower priority node has a free VC. For the situation (a) and (c), select the higher priority node as the output node and for situation (b), select the lower priority node as the output node.

**Remote-congestion-based routing.** If the source node is in the central area, we apply a two-stage routing strategy. In the first stage, calculate the number of free VC of the two candidate nodes, selecting the node with more free VC as the output node. If the two candidates have the same number of free VC, we enter the next stage. In the second stage, weigh the remote hotspot information on the two candidate nodes’ directions. If the weighted values of the two candidates are still the same, select an output node randomly. Otherwise, select the one with little remote congestion as the output node.

Adaptive routing algorithm must consider deadlock freedom. Our proposed ParRouting achieves deadlock-freedom based on Duato theory [27]. We have two VCs (VC0 and VC1) in each router in the network. VC0 has a higher priority than VC1. After VC allocation, VC1 cannot wait for VC0, so as to destroy the cycle dependency between VCs in the network.

### 3.3. Detailed ParRouting Algorithm

ParRouting algorithm is an adaptive algorithm based on minimal routing, therefore, each flit has one or two candidate output nodes as the output node. As mentioned in the last subsection, flit for a “source-destination” pair is first checked its source node’s area in ParRouting. Different routing strategies are used in different areas. The detailed ParRouting for different areas are introduced in the following subsections.

#### 3.3.1. Edge Area

The detailed process is shown in Algorithm 1. For the source node in the edge area, first calculate the candidate output nodes $A_{x}$ and $A_{y}$ by the minimum routing. Then check if $A_{x}$ and $A_{y}$ have a free VC. The node with the higher priority is first checked. If the node has at least one free VC, select the node as the output node, otherwise check if another node has a free VC. If both $A_{x}$ and $A_{y}$ have no free VC, select the node with a higher priority as the output node. If $A_{x}$ and $A_{y}$ have the same priority, check whether the two nodes both have a free VC. If only one candidate has a free VC, selecting the one as the output, otherwise select randomly from $A_{x}$ and $A_{y}$.
**Algorithm 1:** Routing algorithm for nodes in the edge area.
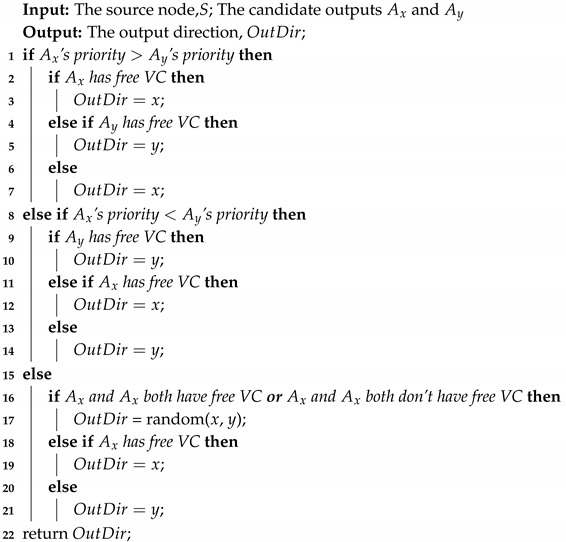


#### 3.3.2. Central Area

For the source node in the central area, we apply the two-stage routing strategy mentioned before. Figure 3 presents a detailed selection strategy for central nodes. A head flit is currently at node S, and its target node is T.

As shown in Figure 3a, $A_{x}$ and $A_{y}$ are the candidate output nodes of S in the $d_{x}$ and $d_{y}$ directions respectively. First, calculate the number of free VC of the two nodes. Select the node with more free VCs to routing if the two numbers have a different number of free VC, otherwise enter the second stage. As shown in Figure 3b, the red nodes $B_{x}$ and $B_{y}$ are the candidate output nodes for $A_{x}$ and $A_{y}$ respectively, their congestion status affect the routing latency directly, so giving them a high weight when calculating the remote congestion. The yellow nodes $C_{x1}$, $C_{x2}$, and $C_{y1}$, $C_{y2}$ are the candidate output nodes of $B_{x}$ and $B_{y}$ in the $d_{x}$ and $d_{y}$ directions respectively. They also have a great influence on routing latency, so give them medium weight when calculating remote congestion. $D_{x}$ and $D_{y}$ in green are common candidate nodes of $C_{x1}$, $C_{x2}$, and $C_{y1}$, $C_{y2}$ respectively. They have a small probability to be used between this “source-destination” pair, so give them a small weight. The weighting method is shown in Algorithm 2.
**Algorithm 2:** Routing algorithm for nodes in the central area.
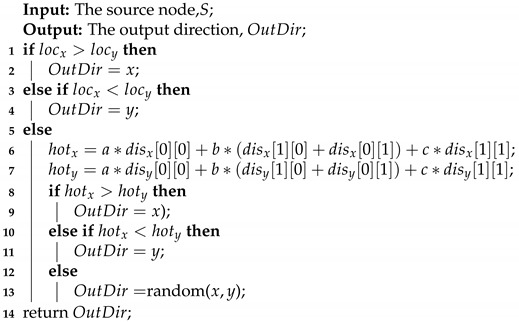


Algorithm 2 shows the detailed routing process for central nodes. The output directions of the two candidate nodes are denoted by $d_{x}$ and $d_{y}$ respectively. As shown in Figure 4, $loc_{x}$ and $loc_{y}$ are used to represent the number of free VC associated with the candidate nodes along the $d_{x}$ and $d_{y}$ directions respectively. $dis_{x}\left[i\right]\left[j\right],\left(i=0,1;j=0,1\right)$ represents the hotspot status of four nodes in the $d_{x}$ direction; $dis_{y}\left[i\right]\left[j\right],\left(i=0,1;j=0,1\right)$ represents the hotspot status of the four nodes in the $d_{y}$ direction.

## 4. ParRouting Implementation

### 4.1. Congestion Propagation

ParRouting leverages the head flits to propagate congestion information in the network without adding a dedicated congestion propagation network. According to Chen et al. [16], in a 128-bit NoC, the head flit of each packet usually has more than 52 free bits Figure 5, which can carry rich congestion information. When the head flit is forwarded to the next node, the node can obtain remote congestion of the coming direction from the flit. Through this mechanism, each node in the network can update the remote congestion information in time.

According to the research of Ma et al. [15], if the number of free buffers of a routing node is less than half, the node can be defined as a hotspot. Therefore, in this paper, the number of occupied virtual channels is used to indicate the congestion status of the node. The hotspot status of a node is marked as 0 when the node is a hotspot, marked as 1 when it is not. After making a routing decision, the hotspot status of the remote node in the opposite direction of propagation is obtained and can be propagated to the next node with the head flit. Before each head flit is sent, the hotspot status of the remote node in the opposite direction of propagation is obtained, and the free bits of the flit is updated before being transmitted to the target node.

In summary, as shown in Figure 6, suppose that node S is going to send a flit to the west neighbor node T. According to the propagation direction, the head flit loads the hotspot status on the east side of S (the node in the dashed box) into its free bits, that is, the status of C and D in the horizontal direction on the east side of S, and their adjacent nodes A, B and E, F in the orthogonal direction. In Figure 6, A and B are hot spots so that the corresponding free bits are recorded as 0. C, D, E, and F are non-hot spots so that the corresponding free bits are recorded as 1. Therefore, for the transmission of S->T in the figure, its free bits value loaded is [0, 0, 1, 1, 1, 1]. When T receives the flit, it updates the hotspot status information of the remote node in the corresponding direction according to the information carried with the flit. Subsequently, the free bits of the flit will be reset to [0, 0, 0, 0, 0, 0] and ready for the next load.

Figure 7 shows the hotspot information available to a node when it receives head flits from all four neighbors. With propagating congestion information in this way, we save the additional congestion-propagation-network wire cost.

According to this mechanism, each node can obtain hotspot information of a remote node in a certain direction after receiving a packet.

### 4.2. The Workflow of ParRouting

The workflow of proposed routing algorithms is shown in Figure 8. The core module is the new added module, regional congestion register (RCR), which is used to record remote node congestion information propagated by head flit. From the beginning, the input port of the router receives a flit. If it is a head flit, RCR will immediately get the hotspot information carried by the head flit and update its knowledge about the hotspot status of the relevant remote links. Then, two candidate outputs are computed. According to the network area of the current node, choose different routing strategies to select the output node. For remote-congestion-based strategy, it requires remote hotspot information from RCR. After selecting the output node, RCR will load remote hotspot information to the head flit’s relevant bits. Thus, that hotspot information can be propagated to neighbor nodes together with the head flit.

The proposed scheme can be integrated into the pipeline of conventional router trivially. The area cost of ParRouting is tiny, which only needs to add 24 bits registers to store remote hotspot information. In addition, compared to traditional regional congestion-aware routing scheme, it has no wiring cost of congestion propagation network. We will evaluate its performance and power consumption in the next section.

## 5. Performance Evaluation

In this section, we evaluate the proposed routing algorithm. The evaluation setup is introduced first, followed by the evaluation methodology, and finally, the performance is analyzed.

### 5.1. Evaluation Setup

We use the gem5 with Garnet2.0 [28] infrastructure to evaluate the proposed routing mechanism. Gem5 provides a cycle-accurate NoC timing model. We use McPat for power consumption statistics, which is widely used for power evaluation.

**Baseline network.** We simulate 4×4 and 8×8 2D Mesh networks respectively. In a 2D Mesh network, each node contains a router and a processing element (PE). Each router has 5 output ports, 4 of which are connected to neighboring routers, one connecting to the local PE. Each router has 2 VCs per port, and 5 flit buffers per VC. The baseline network includes 128-bit link width, minimal routing, wormhole flow control, and Duato’s deadlock avoidance theory.

**Benchmarks.** Both synthetic traffic patterns and application traffic patterns are considered to evaluate the network. Synthetic traffic pattern allows the most generic way to evaluate the network. In this experiment, we use five kinds of synthetic traffic patterns including bit_reverse, bit_rotation, shuffle, transpose, and uniform_random with the mix of 1-flit and 5-flit packet size. On the other hand, application traffic is obtained from real application which results in more realistic scenarios. We use SPLASH-2 to simulate in the syscall-emulation (SE) mode of gem5.

### 5.2. Evaluation Methodology

We compare ParRouting with three typical routing algorithms: XY (a deterministic routing), DyXY (a local congestion-aware routing), and FreeRider (a regional congestion-aware routing). Specifically, we conduct a set of experiments as follows:We first evaluate the overall performance of ParRouting by comparing it with existing routing algorithms on a 4×4 2D Mesh network under synthetic traffic pattern, so as to check if ParRouting can select the output node with high performance for the NoC system;Secondly, we evaluate ParRouting from the perspective of crossbar activity on an 8×8 2D Mesh network under synthetic traffic pattern, so as to verify if ParRouting can balance the traffic loading of the NoC system;Thirdly, we evaluate the overall performance of ParRouting on an 8×8 2D Mesh network under synthetic traffic pattern, to verify the scalability of ParRouting;Lastly, we evaluate the latency of ParRouting on an 8×8 2D Mesh network under SPLASH-2 to verify if ParRouting is effective in realistic scenarios.

### 5.3. Performance Analysis

We compare the performance of ParRouting with XY, DyXY, and FreeRider to evaluate ParRouting. For synthetic traffic patterns, we compare the saturated throughput of the four algorithms. For SPLASH-2, we compare the normalized average latency and power consumption of the four algorithms. The next two subsections introduce the detailed evaluation process for the two traffic patterns.

#### 5.3.1. Synthetic Traffic Pattern

For synthetic traffic patterns, as shown in Figure 9, the X-axis represents the traffic injection rate of the network and the Y-axis represents the average flit latency in different injection rates. To evaluate the throughput of the network, we first evaluate the algorithm in a 4×4 2D Mesh network. We start evaluation from a low traffic injection rate, then increase the injection rate gradually until the network load reaches saturation. To evaluate the scalability of ParRouting, we then evaluate the algorithms in an 8×8 2D Mesh network just like the evaluation method in the 4×4 2D Mesh network. When FreeRider is under the saturated injection rate, we compare the number of crossbar_activity of these algorithms under this injection rate to evaluate the load-balance of the network. The following are the evaluation results.

Figure 9 compares the average packet latency of the four algorithms with increasing injection rate for different traffic patterns on a 4×4 2D Mesh network. As seen in Figure 9, XY is always the first to reach saturated. Considering that we evaluate the saturated throughput, the latency of XY in the high injection rate is meaningless to us. Therefore, we just set the average packet latency up to about 100 cycles. Compared with the classic XY routing algorithm, we can conclude the following experimental results:

(a) In bit_reverse traffic pattern, ParRouting improves the saturated throughput by up to 38.81%; (b) In shuffle traffic pattern, ParRouting improves the saturated throughput by up to 28.72%; (c) In bit_rotation traffic pattern, ParRouting improves by 20%; (d) In transpose traffic pattern, ParRouting improves by 49.95%; (e) In uniform_random traffic pattern, ParRouting improves by 8.7%.

It can be seen that ParRouting shows improvements in all synthetic traffic patterns compared with the XY routing algorithm. As the injection rate increases, the XY algorithm quickly appears congested and the network latency is greatly improved, the throughput of the network goes to saturation, while ParRouting keeps a low latency and transmit packets as normal. This is because, for XY routing, the free buffer in the network does not meet the demand, and the buffer utilization of the network is low. While ParRouting algorithm effectively avoids congestion nodes so that it reduces the possibility of congestion. However compared with FreeRider, ParRouting only improves saturated throughput by 4.37% on average under the 4×4 2D Mesh network. It is because the load-balance function will not work efficiently for a small-scale network. Thus, we compare the algorithms in an 8×8 2D Mesh network.

According to Reza Akbar [18], the load balancing performance of the algorithm can be evaluated by comparing the variance of the crossbar activity in each router. Table 1 lists the mean and variance of the crossbar activity of XY, FreeRider, and ParRouting. Compared with XY and FreeRider when FreeRider is under saturated injection rate. ParRouting reduces the variance of crossbar activity by 28.38% and 33.09% respectively.

Figure 10 shows the crossbar_activity hotspot correspondingly. It can be seen that the central node of XY and FreeRider is more active than ParRouting. This is because we use area-partition routing to achieve a load-balance effect. Compared with other algorithms, ParRouting increases the load on edge nodes and reduces the load on the central node of the network under a higher traffic load. To show the effectiveness of area-partition, we evaluate the performance of each algorithm on an 8×8 2D Mesh network.

Figure 11 compares the average packet latency of each algorithm on an 8×8 2D Mesh network. Similar to the 4×4 network, ParRouting shows better performance than the other three algorithms overall. We compare the saturated throughput of ParRouting and FreeRider especially. Among all the five synthetic traffic patterns, the saturated throughput of ParRouting improves about 8.33% on average compared with FreeRider. The result further proves that ParRouting is an effective load balancing routing algorithm.

#### 5.3.2. Real Application Traffic Pattern

We evaluate the normalized average latency of ParRouting in SE mode under an 8×8 2D Mesh network. Figure 12a,b respectively show the normalized improvement results of the average packet latency and power consumption under the SPLASH-2 benchmark. In terms of NoC latency, ParRouting is generally superior to other routing algorithms. Compared with XY, ParRouting reduces NoC latency in different applications by 13% (fft), 31.3% (lu), 11.4% (lu_non), 44.9% (water), 16% (water-spatial), and 53.4% (ocean). Even for FreeRider, ParRouting performs with a lower average latency of about 5%. In terms of NoC power consumption, ParRouting had less power consumption in all cases compared to its rivals. Compared with XY and FreeRider, the average power consumption of ParRouting in all cases is reduced by 0.641% and 0.318% respectively. Compared with XY, ParRouting had a register in the router, which may lead to more static power consumed by the NoC components. The reason of the trivial power consumption reduction is that ParRouting has lower packet latency.

## 6. Conclusions

This work presents a new area partition-based regional congestion-aware routing algorithm for the NoC system, which aims to balance the network load and reduce network latency. The proposed algorithm divides the network nodes into three priorities and two areas according to th nodes’ closeness centrality. Different strategies were chosen for different areas to make a routing decision. One routing strategy was priority-based and the other was remote-congestion-based. By combining load-balancing and congestion-aware, ParRouting showed a better performance than traditional routing algorithms. On synthetic traffic pattern, ParRouting was 39% better in saturated throughput over XY in the best case and 29% better on average. On SPLASH-2, ParRouting was 53% better in average latency over XY in the best case and 28% better on average.

## Figures and Tables

**Figure 1 micromachines-11-01034-f001:**
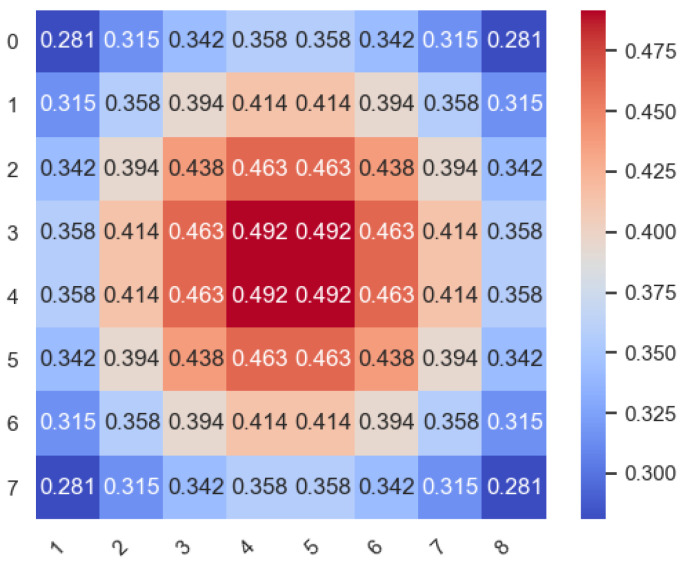
The closeness centrality of an 8×8 2D Mesh network. It is used to partition the priority of the nodes to make routing decisions.

**Figure 2 micromachines-11-01034-f002:**
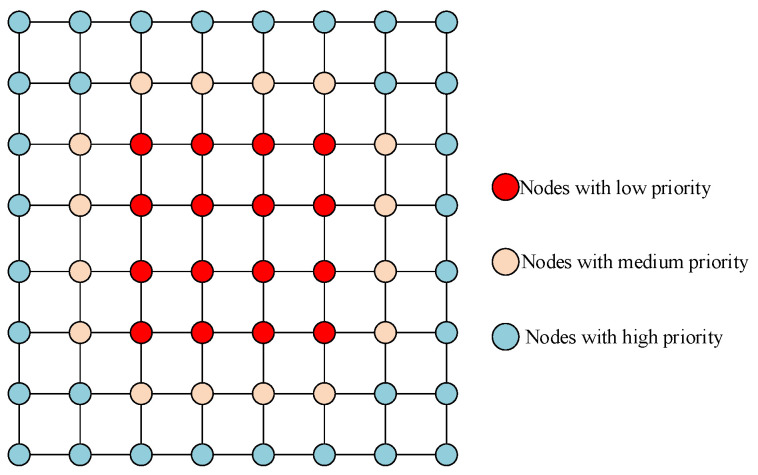
The priority of the nodes in an 8×8 2D Mesh network.

**Figure 3 micromachines-11-01034-f003:**
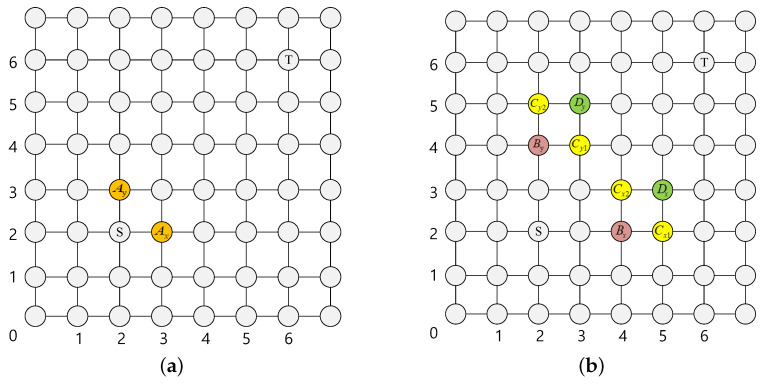
Two-stage routing for nodes in the central area. (**a**) Stage-1. (**b**) Stage-2.

**Figure 4 micromachines-11-01034-f004:**
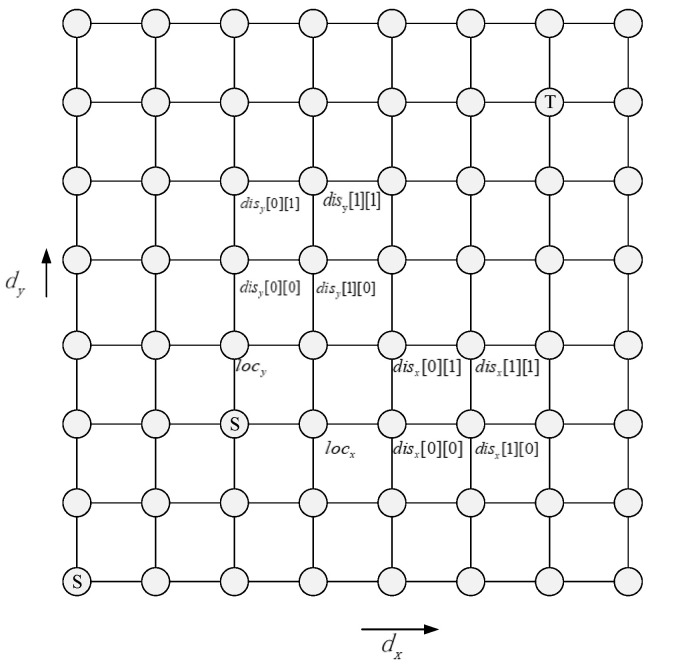
Parameters corresponding to Algorithm 2.

**Figure 5 micromachines-11-01034-f005:**
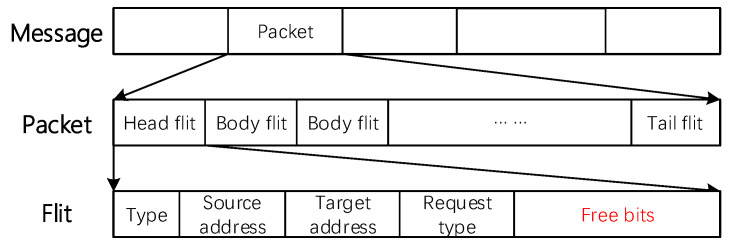
Packet composition.

**Figure 6 micromachines-11-01034-f006:**
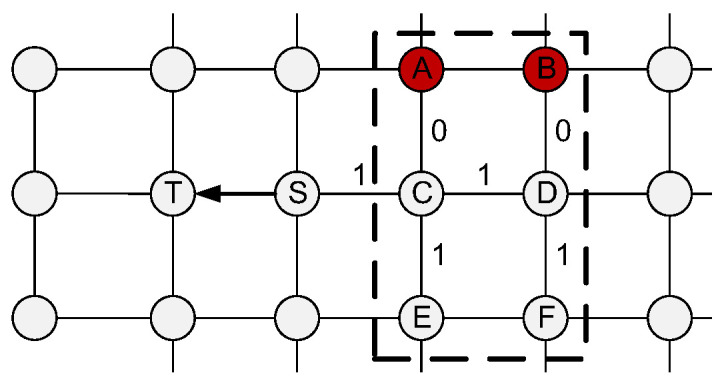
Hotspot information propagated with head flit.

**Figure 7 micromachines-11-01034-f007:**
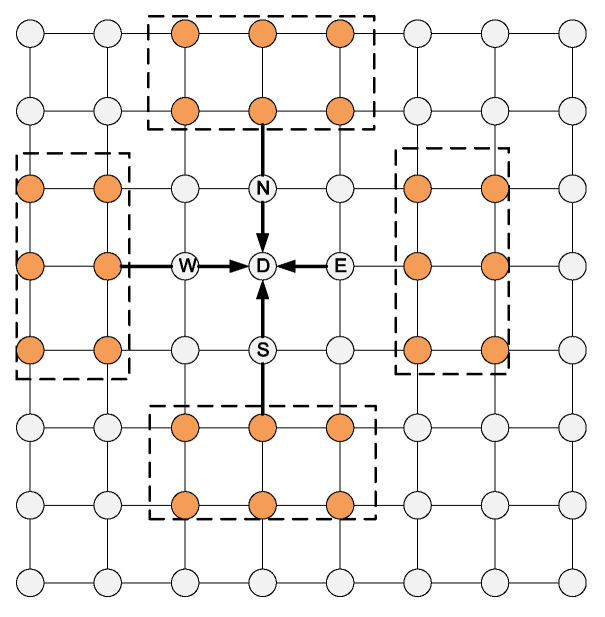
Hotspot information stored by a node.

**Figure 8 micromachines-11-01034-f008:**
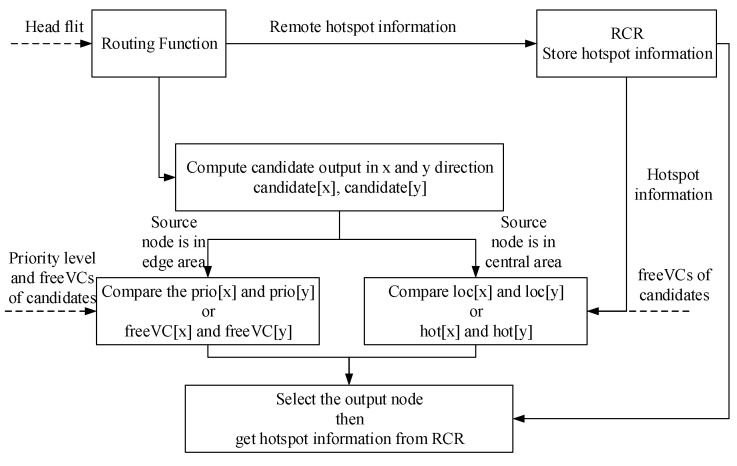
The workflow of ParRouting.

**Figure 9 micromachines-11-01034-f009:**
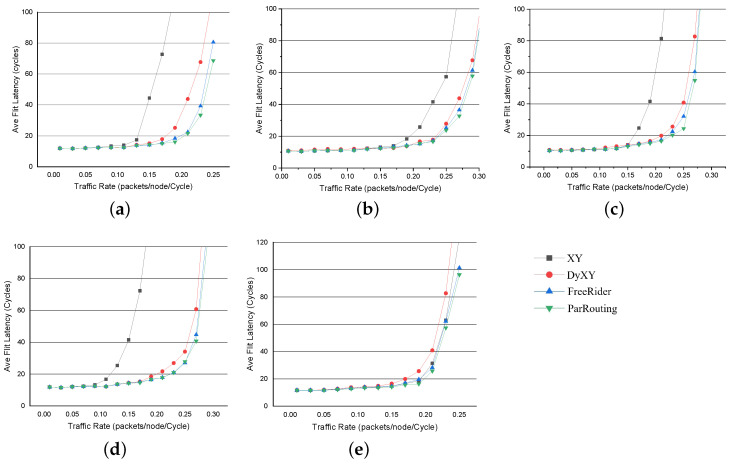
Performance on a 4×4 2D Mesh network for synthetic traffic patterns. For each traffic pattern, the results of the four routing algorithm implementations considered in this work are described. Deterministric routing (XY), local congestion-aware routing (DyXY), regional congestion-aware routing (FreeRider), and our Area Partition-based routing (ParRouting). (**a**) bit_reverse. (**b**) bit_rotation. (**c**) shuffle. (**d**) transpose. (**e**) uniform_random.

**Figure 10 micromachines-11-01034-f010:**
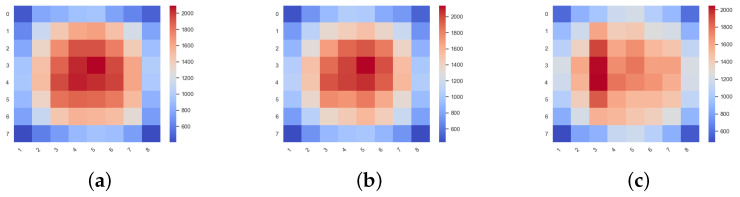
The crossbar activity of each node on an 8×8 2D Mesh network for different routing algorithms. The red to blue represents the crossbar activity from high to low. The more activity a crossbar is, the heavier loading the node has. (**a**) XY. (**b**) FreeRider. (**c**) ParRouting.

**Figure 11 micromachines-11-01034-f011:**
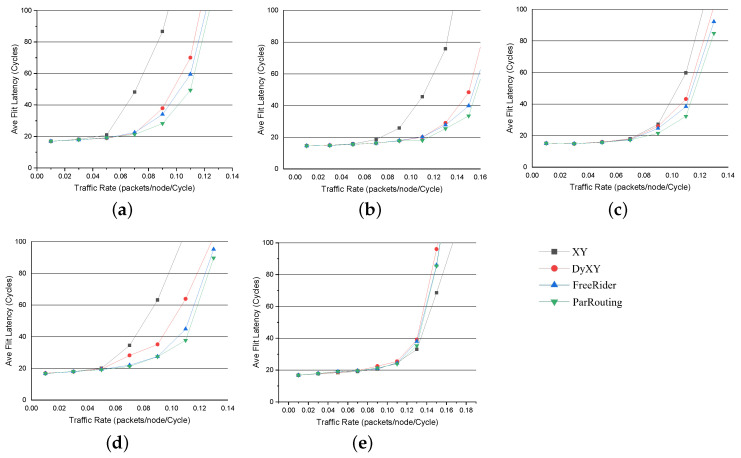
Performance on an 8×8 2D Mesh network for synthetic traffic pattern. (**a**) bit_reverse. (**b**) bit_rotation. (**c**) shuffle. (**d**) transpose. (**e**) uniform_random.

**Figure 12 micromachines-11-01034-f012:**
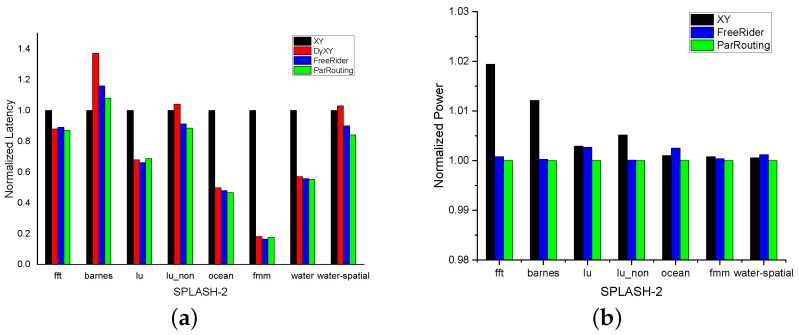
Normalized performance and power consumption for 8×8 2D-Mesh network on SPLASH-2. (**a**) Normalized latency. (**b**) Normalized power consumption.

**Table 1 micromachines-11-01034-t001:** The expected values, variance and variance speed-up of crossbar activity corresponding to Figure 10.

Routing Algorithm	The Expected Value	The Variance	The Variance Speed-Up
XY	1268.265625	494.0542494	1.49446841
FreeRider	1270.234375	461.5427168	1.396124031
ParRouting	1272.234375	330.58862	1

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
