# Peer review of "ParRouting: An Efficient Area Partition-Based Congestion-Aware Routing Algorithm for NoCs"

_micromachines, 2020, doi:10.3390/mi11121034_

Round 1

Reviewer 1 Report

The work introduces a congestion-aware routing algorithm. The manuscript is generally well written/organized and contributions are clearly pointed out.

Authors state that "The earliest local congestion-aware algorithm is RCA, which was proposed by Gratz et al. in 2008" but they also cite as examples of local congestion-aware algorithms DyAD [Hu and Marculescu, 2004] and DyXY [Li et al., 2006].

Authors state also that "the overall performance of NoC systems is still undesirable due to high power cost, long network latency and low network throughput". This is a very generic explanation especially considering it is written in a "Motivation" section. There are many parameters that can be changed in a NoC design to overcome the mentioned problems such as technology and topology.

"It has been proven that assigning 0.7 to k2 would suffice" => Where? Why?

"We assign 0.5 to k1 to calculate threshold1" => Why?

The algorithm is compared with XY, but what about deadlock freedom?

How much the workflow of ParRouting costs in terms of added complexity?

Authors use the concept of closeness centrality in their work. Since this concept is valid with different topologies, what happens to the routing algorithm in topologies different from a mesh?

Please, every time there is an "according to AUTHOR, ..." put also the reference (even if it has been put previously), e.g.:
"According to Akbar, ..."
"According to Chen et al., .."
"According to the research of Ma et al.,"

There are a lot of typos regarding parentheses/punctuation spacing and some regarding the spelling of words (and names), just for example:
"used,so" => "used, so"
dynamic network statues => "dynamic network status"
"technic" => "technique"
"Ebrahmimi" => "Ebrahimi"
"DyXY ,and" => "DyXY, and"
"1)If the two" => "1) If the two"
"a).They" => "a) They"
"b).One" => "b) One"
and so on.. Consider proofreading the manuscript.

When writing about network size please use the "times" sign (×, \times in latex) not the star sign, for example:
8*8 2D mesh network => 8×8 2D mesh network
4*4 network => 4×4 network

Please, expand acronyms at their first occurrence. E.g., VC

Reviewer 2 Report

This  paper examines the efficiency of area partition-based congestion-aware routing algorithm, ParRouting.
This algorithm tries to increase the throughput and to reduce also the output node based on different priorities for higher throughput.
The paper contains extensive introductory and bibliographical material. Also the design and the performance evaluation are well written.
However, I propose an extensive check on layout since the overall presentation could be improved. A good check on format and English language will enhance readability.
I propose ACCEPTANCE WITH MINOR REVISION.

Reviewer 3 Report

I commend the authors for putting this manuscript together. It is well presented. However I have this one suggestion that must be added  to this work before I can recommend its acceptance.

NoC systems suffer from from high power cost, long network latency ,and low network throughput as mentioned in your manuscript. The authors have done justice in showing the performance of ParRouting in terms of latency and network throughput in comparison to other existing routing algorithms.  This makes me wonder, what is the performance of ParRouting  in terms of POWER COST which is pivotal to the performance of NoC systems in general. Thus I strongly suggest the authors show and discuss the power cost performance of their proposed algorithm in comparison to these existing methods.

Round 2

Reviewer 1 Report

The reviewer thanks the authors for their detailed answers.

Consider to add the detailed explanations of responses 3, 6, and 7 in the manuscript to help the interested reader.
For what concerns the application of "Duato's theory", more details are needed since this theory introduces some constraints. Also, a reference to the specific work taken into account shuold be added.

Reviewer 3 Report

I have no other comment

Author Response

Thank you very much for your comments. We proofread the manuscript carefully to check the English and writing format to achieve a better presentation.